# Volatile Content Implications of Increasing Explosivity of the Strombolian Eruptive Style along the Fracture Opening on the NE Villarrica Flank: Minor Eruptive Centers in the Los Nevados Group 2



**Philippe Robidoux [1],*** , **Daniela Pastén [2]**, **Gilles Levresse [3]**, **Gloria Diaz [2]** and **Dante Paredes [2]**

[1] Centro de Excelencia en Geotermia de los Andes (CEGA) y Departamento de Geología, Facultad de Ciencias Físicas y Matemáticas, Universidad de Chile, Plaza Ercilla 803, Santiago 8370450, Chile

[2] Escuela de Geología, Universidad Mayor, Av. Manuel Montt 367, Santiago 7500994, Chile; dmpasten@gmail.com (D.P.); gloria.diazm@mayor.cl (G.D.); dante.paredes@mayor.cl (D.P.)

[3] Centro de Geociencias, UNAM Campus Juriquilla Blvd. Juriquilla 3001, Querétaro 76230, Mexico; glevresse@gmail.com

* Correspondence: robidouxphilippe@gmail.com; Tel.: +56-951232150

**Abstract:** Potential flank eruptions at the presently active Villarrica, Southern Andes Volcanic Zone (33.3–46 °S) require the drawing of a comprehensive scenario of eruptive style dynamics, which partially depends on the degassing process. The case we consider in this study is from the Los Nevados Subgroup 2 (LNG2) and constitutes post-glacial minor eruptive centers (MECs) of basaltic–andesitic and basaltic composition, associated with the northeastern Villarrica flank. Petrological studies of the melt inclusions volatile content in olivine determined the pre-eruptive conditions of the shallow magma feeding system (<249 Mpa saturation pressure, 927–1201 °C). The volatile saturation model on "pressure-dependent" volatile species, measured by Fourier Transform Infrared Microspectrometry (FTIR) ($H_2O$ of 0.4–3.0 wt.% and $CO_2$ of 114–1586 ppm) and electron microprobe (EMP), revealed that fast cooling pyroclasts like vesicular scoria preserve a ~1.5 times larger amount of $CO_2$, S, Cl, and volatile species contained in melt inclusions from primitive olivine ($Fo_{76-86}$). Evidence from geological mapping and drone surveys demonstrated the eruption chronology and spatial changes in eruption style from all the local vents along a N45° corridor. The mechanism by which LNG2 is degassed plays a critical role in increasing the explosivity uphill on the Villarrica flank from volcanic vents in the NE sector (<9 km minimum saturation depth) to the SW sector (<8.1 km), where many crystalline ballistic bombs were expulsed, rather than vesicular and spatter scoria.

**Keywords:** eruptive style; volatiles; Villarrica; melt inclusion; Los Nevados group

## 1. Introduction

The Southern Andes Volcanic Zone (ZVS) extends from 33 to 46 °S [1]. Stratovolcanoes and numerous minor eruptive centers are characteristic of this area. The Los Nevados Group (LNG) adventitious eruptive centers have been defined as a set of volcanic cones and lavas of andesitic–basaltic composition, associated with the northeastern flank of the Villarrica volcano [2,3] (Figure 1a).

Adventitious eruptive centers can share the same stratovolcano magmatic reservoir with which they are associated and are commonly related to the concept of monogenetic volcanism [4]. Although it has been postulated that monogenetic volcanism occurs during a single eruptive cycle, more recent studies ([5], references therein) have indicated that the eruptive history may be more complex. The variability in its formation could indicate changes in magmatic conditions, resulting in modifications in the eruptive style and, consequently, in volcanic hazard conditions, which require attention in the case of the populated area of Pucón (Araucania, Central Chile; Figure 1a).

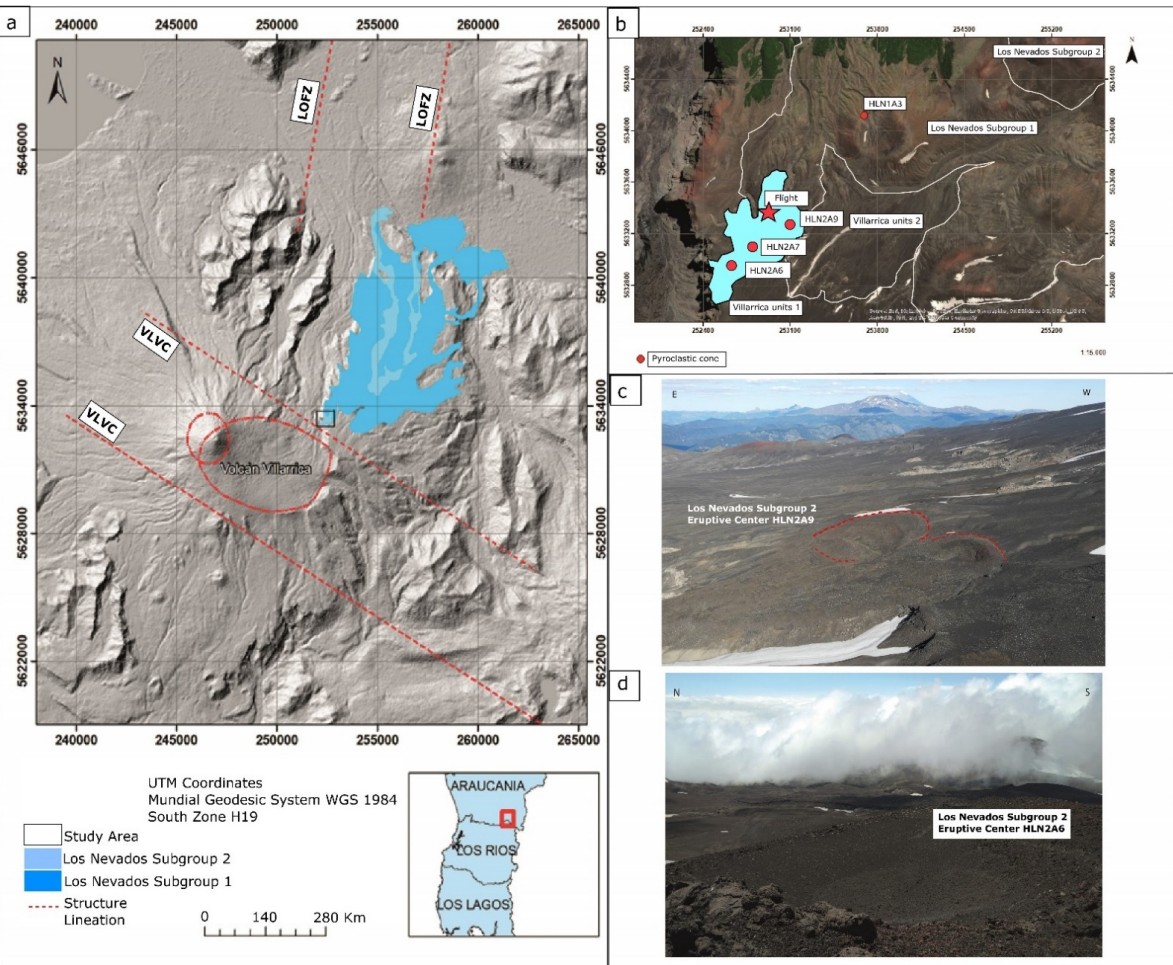

**Figure 1.** Geographical location of Villarrica volcano and Los Nevados Group. (**a**) Location of Villarrica volcano and distribution of the lithological limits from Los Nevados Group (see Legend). The blue polygons in the image correspond to the study area from Los Nevados Group (Subgroups 1 and 2) and the red dotted line to structure lineations adapted from [3] which were redrawn approximatively and projected in ArcMap 9.3 (ArcGIS software from ESRI). Digital elevation model (DEM) of 12.5 m resolution (DEM ALOS-PALSAR; https://vertex.daac.asf.alaska.edu, accessed on 11 December 2018). Structure lineations VLVC stands for Villarrica–Lanín Volcanic Chain and LOFZ for Liquiñe–Ofqui Fault Zone. (**b**) Location of emission centers on satellite view with sampling sites (red dots) and lithological contacts from [3]. The blue polygon is Los Nevados Subgroup 2 from the study area. Drone survey taking off site is represented by the red star. (**c**) Specific views of CEM #HLN2A9 from the Drone taking off site. (**d**) Specific views of CEM #HLN2A6 taken from the ground. Photograph courtesy of A. Sancho.

Scoria or cinder cones are generally the product of moderately explosive eruptive styles, such as Hawaiians, Strombolians, violent Strombolians, and sometimes sub-Plinians [6,7]. The degassing regimes strongly govern eruptive style and building of volcanic cones because the intensity and dynamic of magma ascent is coupled with the characteristics of the system degassing [8]. The pre-eruptive parameters that control the explosive character of such volcanism also include the magmatic ascent rates [9], the magma composition, and the depth of magma transport, among others ([7], references therein).

To understand degassing regimes, the identification of magmatic volatile components and measuring their concentration is crucial, because the system pressure closely depends on the concentration of a gas dissolved in the magma. Given their incidence in pre-eruptive physical processes, magmatic volatiles induce variations on eruptive style, and also the frequency and intensity of magmatic processes [10–12]. Major volatile components that are significant in magmatic systems and preserved in melt inclusions (MIs) include $H_2O$ and $CO_2$, along with S, Cl, and F [12–15].

At Villarrica, the area covered by Los Nevados Subgroup 2 (LNG2) represents the most challenging site to study the relationships between degassing processes and eruptive styles. Despite the distance from the central active craters of Villarrica (5.5 km NE), it is still not clear how those volcanic centers may have been formed on the flanks from the stratovolcano and how their feeding magma differs in composition [16,17]. Pyroclasts and lava cover are observed downhill of the sector of Los Nevados reaching 11 km in length, which is particularly of concern for characterizing the scenario of eventual volcanic activity (Figure 1a). LNG2 represents the series of volcanic cones with the youngest ages from Los Nevados Group (Figure 1b), and their morphologies are well preserved because they represent post-glacial ages [2,3] (Figure 1c,d). Most of all, the volcanic cones follow lineations of seismically active faults such as ZVLO (Figure 1a) [18]. Consequently, the LNG2 could represent a potential natural hazard in the sector between Calafquén, Villarrica, and Caburgua lake.

The access to the study area is facilitated by many local roads, which makes it ideal for repetitive field surveys in comparison to other minor eruptive centers (MEC) on the flanks of Villarrica (Figure 1b). The highest eruptive centers on the topography are of scientific interest (1778 m a.s.l.) because of their location near the limit of the ancient Villarrica Caldera 1 (HLN2A6; Figure 1d). The crustal section of the collapsed caldera border can be easily appreciated from the eastern view of the Villarrica east flank, from the Pichillancahue Glacier (Parque Nacional Villarrica).

On the site, a series of well-preserved parallel volcanic cones cover the reworked volcanic material from Villarrica and LNG1 (Figure 1b). The series of volcanic cones is made up of volcanic products featuring variations in the type of pyroclasts and their textures, implying changes in the eruptive style that are not yet explained in the literature [2,3].

This paper is therefore concerned with two concepts: characterizing eruptive styles and magmatic degassing. The eruptive style was studied according to the textural characteristics of the volcanic deposits and pyroclastic products using optical petrography and scanning electron microscopy (SEM). The magmatic degassing process was studied by determining the volatile components and their concentrations in MIs of olivine phenocrysts. The MIs represent closed capsules that to a certain degree preserve the magmatic volatile contents and chemistry of the pre-eruptive magma composition at the time of their formation [14,15]. The volatile content is measured in MIs of olivine, with electron microprobe analyses (EMPAs) used to determine major oxide concentrations and major volatiles, such as sulfur (S), chlorine (Cl), and fluorine (F). Fourier Transform Infrared Microspectrometry (FTIR) was used to quantify the $H_2O$–$CO_2$ pair for which solubility is strongly dependent on pressure [13]. Finally, a revision in the field helped to draw lines between lithologies over an improved geological map of the sector. The interpreted lithologic successions were correlated to define the chronology of the volcanic activity. This step required building a local digital elevation model (DEM) and basic photogrammetry processing with a drone survey to map the local volcanic structures (Figure 2). This latter task improved the comprehension of spatial changes of eruptive sources and contributed to better explaining the presence of volcanic structures on the Villarrica flanks.

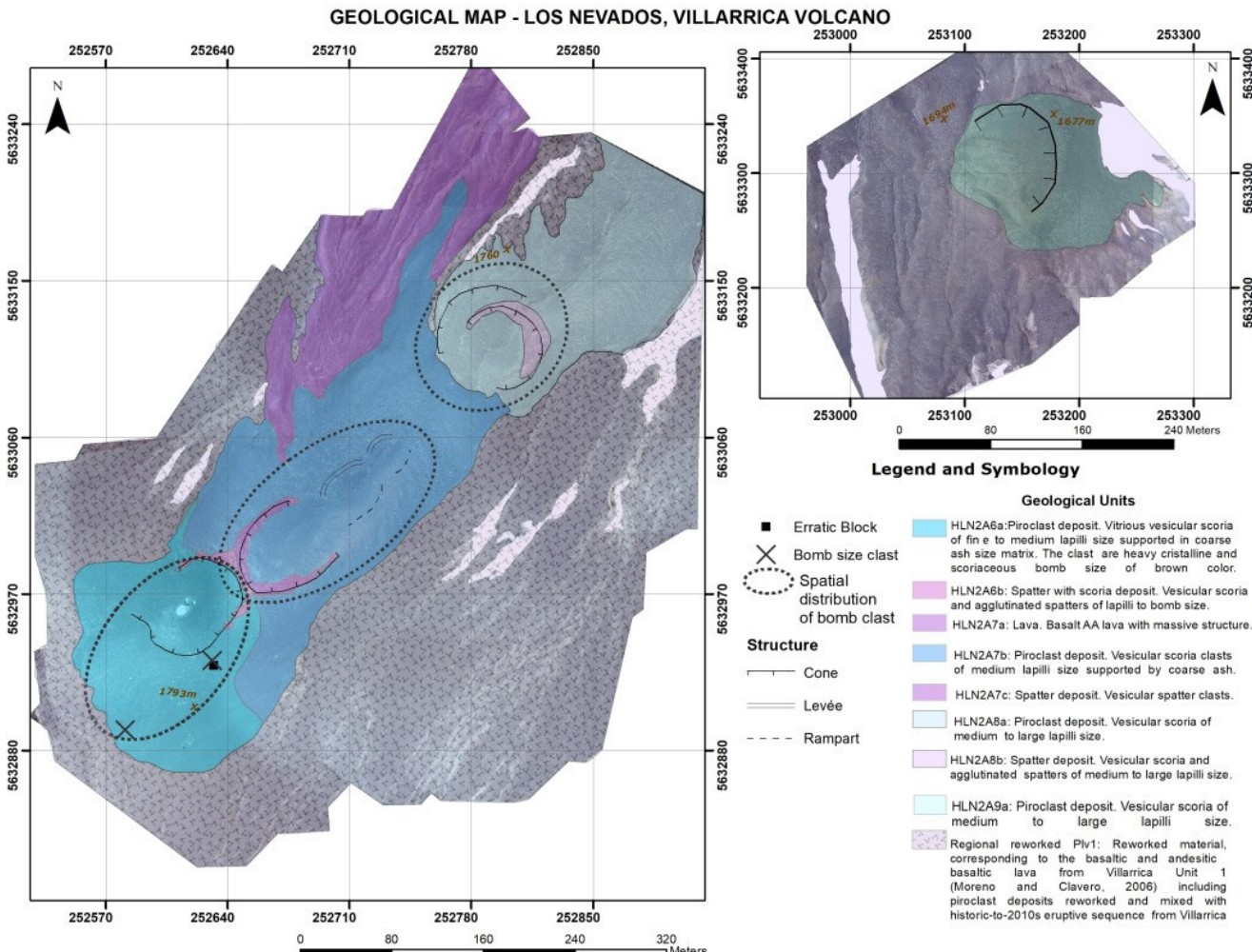

**Figure 2.** Digital elevation model of Los Nevados Subgroup 2. Sector at the highest topographical levels of Los Nevados Subgroup 2 (see Legend). The semi-transparent polygons represent the new lithological contacts between each eruptive center classified between HLN2A6 to HLN2A8. Topographic isoline separated by 2 m were used. Key altitudes points are indicated for lowest and highest sectors. The sector at the top-right corner represents the lowest topographical levels of eruptive vents from the sector, including HLN2A9 volcanic vent.

### 1.1. Geological and Volcanological Setting

This study focuses on the Central Southern Volcanic Zone (CSVZ) where the Los Nevados Group is located. In this active volcanic segment, a ca. 35 km thick crust is characteristic, and the angle of convergence of the Nazca and South America plates is oblique, at about 25° [17,18]. The stratovolcanoes and MECs in the CSVZ produce tholeiitic and high-Al basalt and basaltic andesites [16]. The CEMs of the Los Nevados Group have been classified in previous studies as adventitious volcanoes to the Villarrica volcano [3]. Its eruptive products are mainly of basaltic to andesitic composition (50–58% $SiO_2$), with a predominance of products of andesitic–basaltic composition (52–56% $SiO_2$) [3,19]. Its limit on land corresponds to the eastward extension of the Mocha fracture zone that begins on the Nazca plate [17]. On the continent, Villarrica belongs to the northwestern end of a volcanic chain of northeast–southeast orientation (~N60 °W) of ca. 60 km in length, in which the Cordillera el Mocho, Quetrupillán, Quinquilil, and Lanin volcanoes are also located [17]. Of these, only the Lanin, Quetrupillán, and Villarrica volcanoes erupted recently during the Holocene [3]. Volcanism in the CSVZ is spatially related to the Liquiñe Ofqui Fault Zone (LOFZ). The LOFZ represents the main structure that controls tectonic activity in the area and has an approximate length of 1200 km.

The Villarrica volcano is a Middle Pleistocene to Holocene eruptive center located at 39°24′ S and 71°55′ W (Figure 1a). It is mainly made up of basaltic to andesitic–basaltic lavas, lahars, pyroclastic flows, and fall deposits. Its basement is restricted to rocks of Devonian–Pleistocene age and comprises the Trafún Metamorphic Complex, sedimentary sequences that make up the Panguipulli Formation, and volcanoclastic sequences belonging to the Pino Huacho Stratum, the Pucón Península Strata, and the Huincacara Strata [2]. The history associated with the development of the Villarrica itself is grouped into three associated evolutionary stages based on stratigraphic, geomorphological, structural, and geochronological criteria, forming Villarrica units 1, 2, and 3 [2,3].

The Holocene-to-present volcanic activity includes ~49 historic eruptions that have occurred at Villarrica since 1558 [20–23]. The 1948–1971 sequences are particularly well described in terms of the pre-eruptive conditions [24–26], and there is also preliminary and detailed information on the magmatic volatile content from Chaimilla tephra produced by the explosive eruptions [27,28]. Since the post-glacial period, Villarrica has demonstrated an effusive and weak Strombolian eruptive style at the central crater, which is also found at the corresponding adventitious cones [2,3].

Central volcanic activity at Villarrica includes the adventitious vents (Chaillupén, Los Nevados); these are centers of interest, in addition to regional minor eruptive centers and the possible connection to the monogenetic field from the Caburgua–Huelemolle Small Eruptive Center (CHSEC). The latter may also involve a larger area of potential seismic/volcanic hazards [18,19,25,29,30].

### 1.2. Local Setting: Los Nevados Group

The Los Nevados Group is a set of volcanic cones and associated lavas of andesitic–basaltic composition of the Holocene age (<11,700-year B.P.) located on the northeast flank of the Villarrica volcano, northeast of the edge of Caldera 1 (Figure 1b). They are controlled by a main fissure of orientation N60 °E [3] and divided into two subgroups (Figure 1a,b) according to their spatial arrangement and age [2,3]. Los Nevados Subgroup 1 (LNG1) (Late Holocene, >2600 years B.P.; $^{14}$C in carbonized wood resumed in [4]) is a set of volcanic cones that were generated from four fissures through which the magma would have reached the surface. These cones have a subcircular base of up to 800 m in diameter, heights of less than 80 m, and craters with diameters of up to 350 m, which are generally open [3]. The age of these deposits is limited stratigraphically by the fact that this subgroup is partially covered by pyroclastic deposits from Villarrica Unit 3, deposit reported as 2.620 ± 210-year B.P. and measured by Moreno (1994a). This indicates that LNG1 formation would have occurred prior to that age [2,3].

Los Nevados Subgroup 2 (LNG2) (Late Holocene, <2600 years B.P.; Ibid. to reference material for LNG1) is a set of volcanic cones and lavas of andesitic–basaltic composition, generated from two of four fissures through which the magma would have reached the surface. These volcanic cones have a subcircular to elliptical base of up to 600 m in diameter, with heights that are less than 60 m and craters with diameters of up to 200 m, which are generally horseshoe-shaped [3]. The minimum age of these deposits is not stratigraphically controlled because this subgroup is not covered by pyroclastic deposits of Villarrica Unit 3. Therefore, it is estimated that their formation would have occurred less than 2600 years B.P. [2,3]. The sector on the highest topographic level corresponds to a series of scoria, spatter, and non-vesicular bomb size-formed cones, oriented along a N45 °E direction (Figure 1b–d).

## 2. Materials and Methods

### 2.1. Field Work

The field work for this study consisted of three repeated field campaigns of 4–5 days duration each (2017–2020) to obtain stratigraphic information and take samples of rock fragments for subsequent analysis. Petrographic characteristics and volatile contents were investigated from juvenile pyroclasts collected at significant levels from the falling deposits

of three volcanic cones, starting from the highest ground level in sector SW to the limit NE (HLN2A6, HLN2A7, and HLN2A9; Figure 1b–d). All these edifices were included within the Los Nevados Subgroup 2 (LGN2), corresponding to the last eruptive event associated with the formation of CEMs in the Los Nevados Group. The stratigraphic sequences of the eruptive centers are described in the original work of Pastén [31] (2018), and such sequences with corresponding pictures of sampled pyroclasts can be found in the Supplementary Materials (Figure S1).

### 2.2. Unmanned Aerial Vehicle (UAV) Survey

The recognition of volcanic deposits is shown in Figure 2. All details of field textures were interpreted with acquisition of orthomosaic photography, taken by a light Unmanned Aerial Vehicle (UAV) engine Mavic Air 2 (manufactured by DJI Innovations: http://www.dji.com, accessed on 10 March 2021). Instead of using it for monitoring (i.e., [32,33]), the UAV was brought into the field to create a high-resolution digital elevation model (DEM), a task requiring that homogeneous resolution images (4000 × 3000 pixels, horizontal resolution = 72 ppp, vertical resolution = 72 ppp) be acquired directly from the engine camera (ISO 110, focal distance of 4 mm, time exposure of 1/320S, color RGB). The processing of $n = 30$ images for the 1000 × 500 m dimension sector was made possible with the Structure from Motion (SfM) software Pix4UAV [34,35], which is a well-developed technique for far remote areas [36]. Twenty-four repeated flights at 50 m from the ground were performed at 12 ground control points (GCPs) (including the starting point in Figure 1b) under weak wind conditions (<30 km/h), with coordinates taken with a Garmin eTrex GPS to verify each x–y–z position from the flight was at the correct position (1694–1794 m a.s.l.). The field campaign lasted for two days, 7–8 February 2021, using a drone battery of 11.55 Ah, which allowed the quadcopter to fly for about 30 min, six times (3 h), without snow cover, during the ideal period of the year for clear weather (the dry summer season in the Southern Hemisphere).

### 2.3. Petrography

Petrographic observations were performed on a total of four thin sections produced in the laboratory belonging to Mr. Rubén Espinoza (Santiago, Chile) using the most representative sampled pyroclasts in the field (Figure S1). The samples were clasts contained in the deposits of the eruptive vents HLN2A6, HLN2A7, and HLN2A9. These were selected and characterized both mineralogically and texturally using a petrographic microscope while crystal separates from olivines described for their textures. The mineral abundance estimates were found through the modal counting method. More information regarding the mineral phases, textures, sizes, and abundances can be found in the Supplementary Materials (Table S1). Bulk rock analyses were realized on the corresponding samples described for petrographic characteristics (Table S2) according to the preparation procedure from Activation Laboratory (Ancaster, ON, Canada). The method employs a lithium metaborate/tetraborate fusion for measuring major as trace elements by Fusion Inducted Coupled Plasma (ICP) with Optical Emission Spectrometry (OES) and Mass Spectrometry (MS).

### 2.4. Scanning Electron Microscope

Four scoriaceous fragments were selected from a granulometric fraction of ~1–2 mm that was encountered within the bomb and lapilli-sized clasts collected on site. The selection of the smallest clasts allowed us to observe the groundmass texture not visible in the petrographic sections, but no phenocryst-free rocks of this size were encountered among clasts smaller than the lapilli-size formats (Figure S1). The observation was undertaken in the Crustal Fluids Laboratory of the Geosciences Center (UNAM) in Juriquilla, Mexico. For this, a Hitachi TM-1000 Scanning Electron Microscope (SEM) was used, with a backscattered electron detector and an Oxford energy dispersive spectroscopic (EDS) device coupled to an acceleration voltage of 15 kV.

A semiquantitative X-ray microanalysis was performed by EDS, with an acquisition time of 60 s. BSE (backscattered electron) images were obtained with magnifications of 80×, 150×, and 300×. The images obtained by this technique are provided in the Supplementary Materials (Figure S2).

### 2.5. Populations of Olivine

Small-to-medium lapilli size fragments were crushed for sieving to obtain a 1–2 mm fraction and used to collect mafic phenocrysts (olivine, pyroxene), at the Juan Vargas Laboratory in Santiago. From a population of 100 olivine crystals belonging to each eruptive center studied (HLN2A9, HLN2A7, and HLN2A6), descriptive statistics show the main textural characteristics that represent each of them, according to the methodology proposed by Robidoux [37]: (G1) olivine without inclusion, (G2) olivine with closed inclusion, (G3) olivine with closed inclusion and bubble, (G4) olivine with heterogeneous bubble (oxides + bubbles), (G5) olivine with crystallized inclusion, and (G6) other (referring to re-entrant, hourglass, or mineral phase inclusions in the inclusion). In turn, descriptive statistics were obtained for the samples that were analyzed by FTIR by using a model SMZ171Motic binocular microscope with wide-field objectives (up to 10× magnification). The textural and morphological statistical analyses showed the magmatic conditions and reflected degrees of isolation with respect to the coexisting melt [37,38]. For more detail, see the Supplementary Materials (Figures S3–S5; Table S2).

### 2.6. Fourier Transform Infrared Microspectrometry (FTIR)

The content of $H_2O$ and $CO_2$ in the MIs was measured using the FTIR technique, with a Hyperion Bruker FTIR spectrometer interconnected with a Continuum IR microscope, belonging to the Laboratory of Crustal Fluids of the Geosciences Center of the Juriquilla Campus, located at the National Autonomous University of Mexico. In total, $n = 81$ analyses were performed on the inclusions with glassy texture. Details on the Microanalysis of melt inclusions, preparation, and data processing are described in Figures S6 and S7.

Through the relationship between the concentration among the Beer–Lambert Law (Equation (1)), the content (C) of $H_2O$ and $CO_2$ was calculated.

$$C = \frac{MA}{\delta \rho E} \tag{1}$$

The numerator parameters used to carry out the calculations in Equation (1) correspond to the molar mass of the analyte (M) (g/mol) and the absorbance (A). The denominators are the thickness of the wafer ($\delta$) (in millimeter for calculation, but in micrometer for data reports), the density of the melt inclusions ($\rho$) (g/cm$^3$) and the molar absorption coefficient $\varepsilon$ (L/mol).

For the case of water, the total concentration (C) of $H_2O$, dissolved as molecular water, was determined from the absorbance bands at 3550 cm$^{-1}$ [39]. The absorbance (A) was measured in the OPUS program, with correction for the baseline. The parameter $\delta$, $\rho$, and $\varepsilon$ are determined individually for each inclusion and explained in detail with Figure S7 (see also, e.g. [39–43]).

Measurement was carried out wherever the characteristic vibrations of the $CO_3^{-2}$ species were present [44,45]. The absorbance peaks of the spectrum corresponded to the wavelengths of 1515 and 1435 cm$^{-1}$, and sometimes 1350 cm$^{-1}$ being detected [41,44]. Baseline correction and the peak fitting model were applied using the degassed glass spectrum background [46]. The molar absorption coefficient ($\varepsilon$) for the bands on which the carbonate concentration was measured for basaltic glasses were compositionally dependent [47], for which we used the molar average of Na/(Na + Ca), as follows:

$$\varepsilon = 451 - \left[ \frac{Na}{Na + Ca} \right]. \tag{2}$$

The abbreviations used for the calculations in Equation (2) correspond to molecular fraction of sodium (Na) and calcium (Ca), which are determined for the composition of each inclusion according to the linear equation reported in Dixon work [47].

*2.7. Electron Microprobe Analyzer (EMPA)*

Inclusions, interstitial glasses, and mineral chemistry were probed with an electron microprobe analyzer (EMPA), model JXA-8200 (manufactured JEOL in Tokyo, Japan), equipped with five wavelength-dispersive X-ray spectrometers and one energy-dispersive X-ray spectrometer analyzer, at the HPHT (high-pressure/high-temperature) laboratory of the Istituto Nazionale di Geofisica e Vulcanologia (INGV) in Rome (Italy). All crystals were prepared on different mounts after using abrasives and diamond polishing powder fractions (down to 6, 3, and 1 μm), and then finished with 0.1 micron aluminum oxide. In Table S4, duplicate spots were performed on two secondary reference glass materials (GL07 D30-1 and ALV519-4-1 [48]), measured two or three times per day during the two-day analytical period. Reproducibility of results from major elements using the standard deviation rhyolitic glass GL07 D30-1 (0.13 S.D.%) and basaltic glass ALV519-4-1 (0.06 S.D.%) set of reference crystals was used for quantifying each major element's precision. Matrix glass on the rim of separated phenocrysts and background (matrix) glass obtained from polished phenocrysts were placed over indium mounts and were probed by EMPA equally to the glass surfaces of inclusions and phenocrysts, before being pressed for 24 h so they were parallel on the mounts. For both the glass and mineral surfaces, the EMPA conditions were a beam current of 7.50 nA, accelerating voltage of 15 kV, and beam diameter of 5 μm. The counting times for the minerals were 10 and 5 s at the peak and background, respectively.

## 3. Results

*3.1. Petrographic Characteristics of the Deposits*

3.1.1. Deposit HLN2a6

- This is a double-nested crater with the SW side cutting off the NE side. A deeper base is observed on the NE crater, where a solid gray-black to brown overflow rampart deposit consists of unconsolidated-to-agglutinated scoria with a minimum thickness of 9 m (Figure 2 and Figure S1; HLN2A6a and HLN2A6b). The base of the sequence consists of vitreous vesicular scoria with a medium lapilli size (<~20 cm), and a lesser quantity of fine lapilli (30% of the basal sequence), immersed in a 10% coarse ash-sized matrix (Figure 1d and Figure S1a). Toward the top of the deposit, the size of the clasts increases, consisting essentially of juvenile scoriaceous fragments of a block size of ~60 cm in diameter, with predominantly brown tones (70% of the upper sequence), containing two groups of pyroclasts (HLN2A6A and HLN2A6F). The SW crater is covered with the rest of the series and forms a semiarc rampart structure with a cumulation of bombs (lapilli to block size) dispersed as far as 200 m from the crater (Figure 2). Lapilli fragments were sampled such as crystalline and heavy block HLN2A6 and fusiform scoriaceous HLN2A6F (Figure S1).
- The first group (HLN2A6A) of lapilli is characterized by being hypocrystalline and porphyric, and it presents variations in vesicularity. The pyroclastic fragment obtained from the second group is vesicular with a fusiform morphology (HLN2A6F), characterized by being hypocrystalline, porphyritic, and moderately vesicular (Figure S1 and Table S1). The phenocrysts (5 vol% of the total rock) correspond to plagioclase (4 vol%, 1–2 mm) of subhedral form and olivine (1 vol%, ≤1 mm) of greenish tones and subhedral form. The groundmass (60% of the total rock) is microcrystalline (Figures S2 and S4).

3.1.2. Deposit HLN2a7

- This is an overflow levee crater with AA blocks and lapilli clasts (Figure 2 and Figure S1). The sequence appears as a massive gray-brown deposit

of approximately 3 m thickness, made of agglutinated scoria (spatter texture). This eruptive center is made up of abundant highly vesicular scoriaceous juvenile clasts of medium-thick lapilli size (10% of the deposit), which exhibit black-brown colorations and clasts (80% of the deposit) of bomb size with diameters of ~30 cm, which are arranged in an aggregate manner and are immersed in a matrix of coarse ash size (10% of the deposit) (Figures S2 and S4 and Table S1).

- Petrographically, the spatter clasts are basaltic vesicular olivine, mostly hypocrystalline and porphyritic, and the rest of the groundmass is highly vesicular (Figure S4, Table S1). The sampled fragment is made up of phenocrysts (3 vol% with respect to the total rock) of subhedral plagioclase (2%, 0.5–1 mm) and subhedral olivine (<1 vol%, <1 mm). The groundmass (67% of the total rock) has an aphanitic texture. Vesicles occupy from 30 vol% and up to 50–60 vol% of the total rock, and are essentially <1 mm, decreasing in size towards the center of the sample (Figure S1). At microscopic level and using a scanning electron microscope (SEM), the clasts are revealed to be vesicular basalt containing interstitial accessory clinopyroxene (2 vol%) and olivine (<1 vol%) among dominant 5 vol% plagioclase. The rock is highly vesicular (~60 vol% of the total rock) and has a hypocrystalline and vitrophyric texture (Figures S2 and S4).

### 3.1.3. Deposit HLN2a9

- This is a circular crater with an excavated 5.4 m steep deposit of scoria on the north side contrasting with the rounded morphology that reaches 12 m high on the south side (Figures 1 and 2 and Figure S1). The sampled north wall is a solid gray-black weakly consolidated deposit made up of vesicular scoriaceous clasts of medium-thick lapilli (90%) immersed in a matrix of coarse ash (10%). Gradually, at higher levels, the size of the pyroclastic fragments increases until it is made up of bomb size (~20%) of up to 30 cm in diameter at their greatest length, and which are scoriaceous and highly vesicular.
- Microscope photos show a rock that represents an interstitial vesicular groundmass of clinopyroxene and olivine. This rock is highly vesicular (58 vol% of the total rock), hypocrystalline, and vesicular and porphyritic in texture. The phenocrysts make up 6 vol% of the total rock and include phenocrysts of plagioclase (4 vol% with respect to the total rock, 1–3 mm), which locally exhibit sieved textures and partially resorbed edges (Figure S4); olivine phenocrysts of subhedral form (1 vol%, 1–2.5 mm); olivine microphenocrysts of subhedral form (1 vol%, 0.1–0.6 mm); and subhedral clinopyroxene microphenocrysts (<1 vol%, 0.2–0.4 mm). The groundmass (36 vol% of the total rock) has an interstitial texture and is made up of plagioclase microlites (6 vol%, <0.5 mm) partially surrounded by pyroxene (Figure S4, Table S1).

### 3.2. Bulk Rock Chemistry

Whole-rock compositions are defined as basaltic andesites ($SiO_2$ = 54.5–55.2 wt.% and $K_2O$ = 0.78–1.02 wt.%) and plotted in the field from calc-alkaline (CA) to high-potassium calc-alkaline (HKCA) rocks (Figure 3), while the total alkalis of the present samples varied from 4.1 to 4.5 wt.%. The most primitive sample was the HLN2A9, with 54.5 wt.% $SiO_2$ and MgO = 4.46 wt.% (Table S2), with relatively low $K_2O$ (0.78 wt.%). This scoriaceous bomb reached Mg# = 0.44 (ratio of molecular weighted magnesium content to total magnesium+iron as total iron as $Fe^{2+}$). Major oxides from LNG2 falls within the range of Villarrica bulk rock results available in the literature (51.0–59.0 wt.% $SiO_2$ and 0.44–1.08 wt.% $K_2O$ [27,28,49–51]). It was slightly more differentiated than most minor eruptive centers (MEC) (50.0–52.5 wt.% $SiO_2$ and 0.35–1.08 wt.% $K_2O$ [25,29,30]).

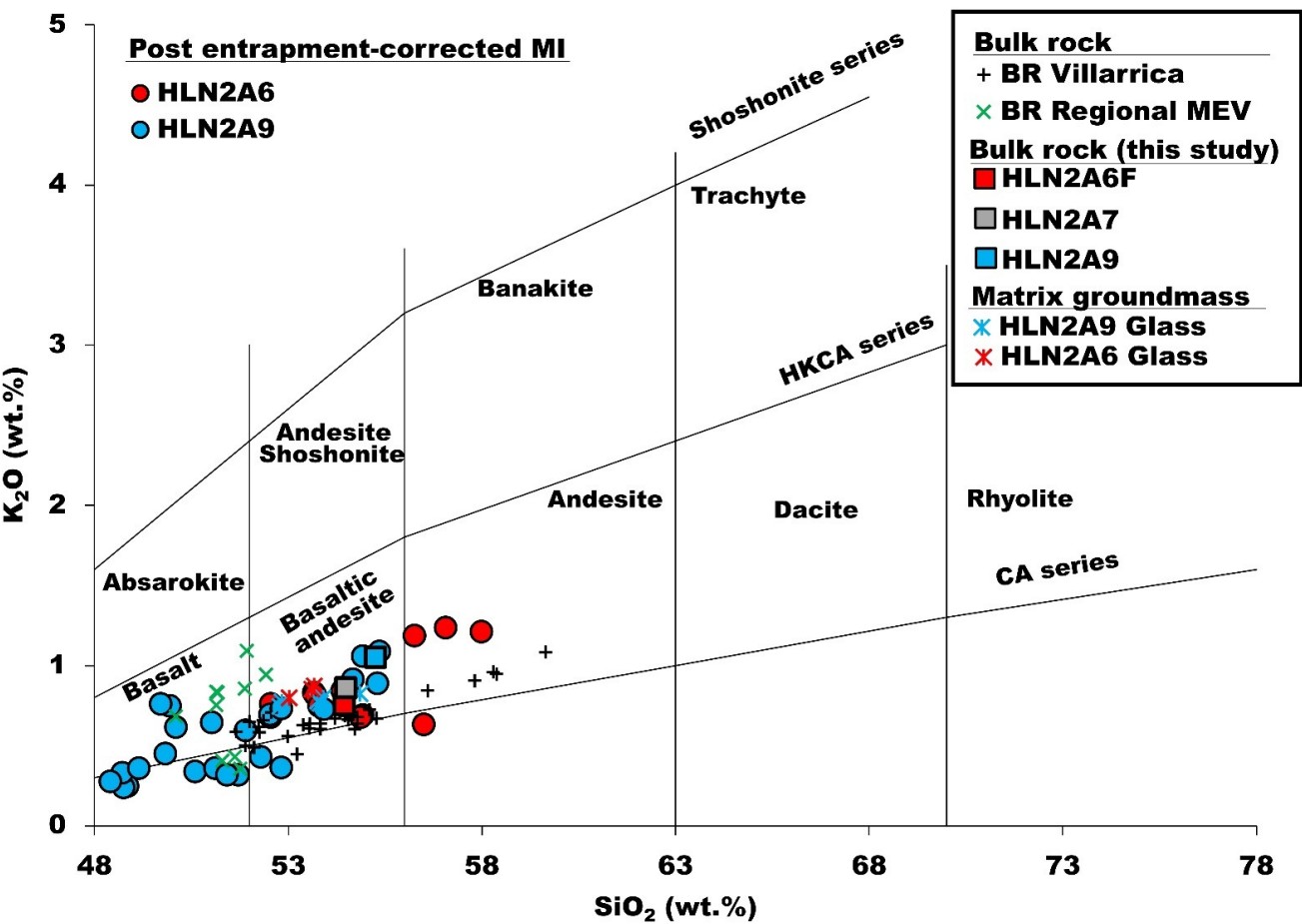

**Figure 3.** Major element composition of volcanic products from Los Nevados Subgroup 2. Post entrapment-corrected MI with their $K_2O$-vs-$SiO_2$ classification. Basalt and basaltic andesite phenocrysts-hosted MI are shown as rounded shapes with red (HLN2A6) and blue (HLN2A9) filled color. Bulk rock samples from Los Nevados are represented by the colored squares. Repeated analyses involving international standard glasses resulted in errors of ≤1%. Literature data from Villarrica are represented by black cross, while MEV from regional sector close to Villarrica are illustrated with green X symbol (See references in Supplementary Material Figure S6).

### 3.3. Olivine Populations

The G2 and G3 categories are of greatest interest for carrying out volatile content measurements. The results indicated that the highest percentages of MI of the mentioned categories occurred in HLN2A9 (21%) and HLN2A6F (19%). The samples from the other eruptive centers, HLN2A7 (10% G2+G3) and HLN2A6A (1% G2+G3), showed a predominance of crystals with completely crystallized inclusions (~>70%), or simply crystals without inclusions (G1) (Figure S5).

Selected inclusions for analyses (FTIR, EMPA) were attributed textural characteristics, for a total of *n* = 42 samples (Table S3, Figure S5). Groundmass glass in contact with phenocrysts (*n* = 4) was used as the standard for the absorbance spectrum in the FTIR data treatment. Among the analyzed inclusions, a ratio of 9/42 was found for closed glassy texture inclusions (G2), 23/42 contained a shrinkage bubble (G3) [52] (i.e., [52,53]), 3/42 contained a shrinkage bubble with possible secondary phases, and 7/42 were reentrant or hourglass (connected to the border of the crystal). The single phenocryst sample HLN2A91 included 12/42 analyses for which the results were repeated for testing the variability of chemical characteristics between MIs in different areas of the sample (Figure 4).

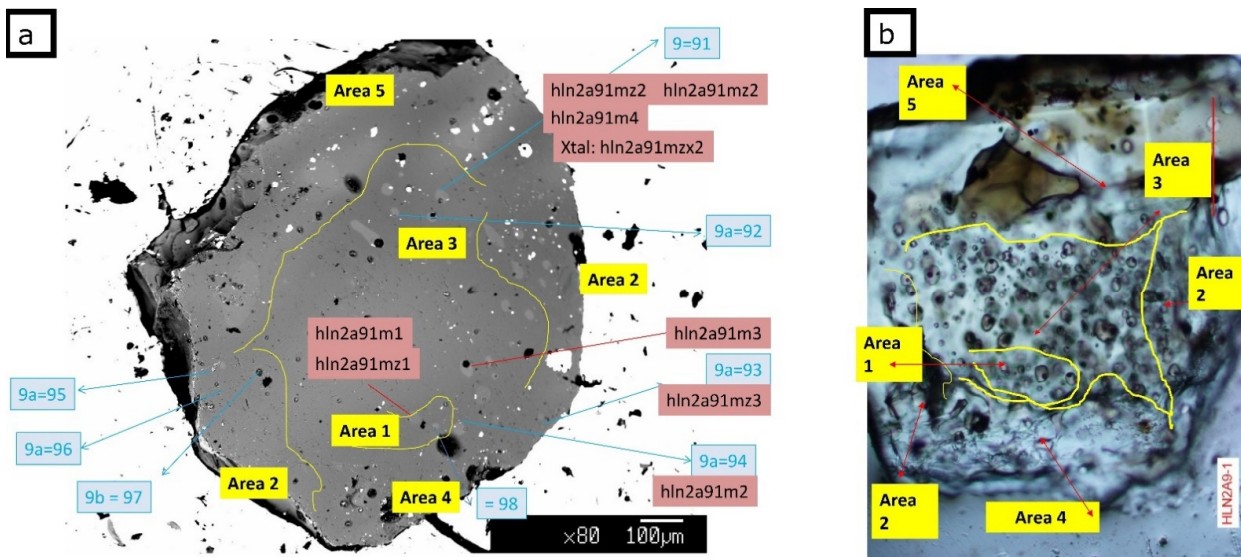

**Figure 4.** Sample HLN2A91 and analytical spots. (**a**) Electron microprobe image with backscattered electron image (BEI) showing intersected inclusions on crystal HLN2A91. In total five different sectors are showing different kind of inclusion textures for which we attribute the letter "A ". Red rectangles with black letters correspond to EMPA results while blue rectangles with blue letters to FTIR results. (**b**) Microscope image of sample HLN2A91 with different sectors. Area 1 (A1): Spherical glassy inclusion with bubbles (still not classified as typical shrinkage bubble), but also without bubbles (best preserved). Area 2 (A2): Irregular elongated glassy inclusion with bubbles. Area 3 (A3): Spherical glassy inclusion with bubbles. Area 4 (A4): Glassy transparent elongated inclusions. Irregular closed and open (reentrant) inclusions. Isolated dark cubic/angular opaque solid oxide phases. Area5 (A5): Glassy coffee translucid elongated inclusions. Irregular closed and open (reentrant) inclusions. Additionally, they contain cubic/angular dark coffee color and black opaque solid oxide phases.

### 3.4. Infrared Microscopy

The OH- peak was clearly detected in $n = 41$ samples [43,54]. The water concentrations are reported in Table S5 including graphical precision, for which the total $H_2O$ peak at $3550$ cm$^{-1}$ corresponded to the fundamental OH- stretching vibration [39,54]. Replicate precision (1σ) was tested on $n = 8$ samples with $\pm0.098$ wt.% [$H_2O$] variation with error of 14.5%. The reproducibility of the measurements was tested as double spots on the same inclusion and the % error for 1σ propagated systematically for all water contents (Tables S3 and S5). The density is not considered as source of error as it depends on the inclusion major oxide contents (Table S5: range between 2.49 and 2.64 g/cm$^3$). Samples with clean infrared reflective signals were used for the interference fringes method [55]. The thicknesses determined by this method agreed to within an average of 7 μm (% error of 30.7% in water content), in comparison to direct thickness measurement errors [43]. The ranges of water contents in LNG2 were 0.42–2.77 ($1.07 \pm 0.96$ wt.%) in HLN2A6 and 0.8–2.99 ($1.15 \pm 1.05$ wt.%) in HLN2A9. The series of inclusions observed in HLÑN2A7 were mainly crystallized, and the only result obtained was 0.48 wt.% $H_2O$. The sample HLN2A91, in total obtained $n = 10$ valid results between 0.86 and 2.99 (mean $1.79 \pm 0.80$ wt.% $H_2O$). The specimen is used for comparing the variability of water content between different types of MIs (with or without shrinkage bubbles) and observed textural assemblages in distinct areas (A1, A2, etc.). The measured range is not related to an analytical issue, but seems to reflect natural processes.

A double peak of $CO_3{}^{2-}$ was detected in $n = 17$ samples, with average 1σ in $CO_3{}^{2-}$ of $\pm175$ ppm and an average percent error of 35.7%. Only five of these samples were successfully corrected for the doublet anomaly (as in [46]). This ultimate method yielded an average of 1σ in $CO_3{}^{2-}$ of $\pm298$ ppm and an average percent error of 39.0% ($n = 5$). The range of $CO_3{}^{2-}$ contents in LNG2 is 115–1069 ($484 \pm 388$ ppm) in HLN2A6 ($n = 5$) and 98–1586 ($610 \pm 578$ ppm) in HLN2A9 ($n = 12$). In Beer–Lambert's Law, for calculating the

carbon concentrations, the values of the absorption coefficient ranged between 388 and 415 L/mol·cm (Table S6), depending on the inclusions [47].

### 3.5. Melt Inclusions and Matrix Glass Composition

All the MI compositions illustrated in Figure 3 were corrected for post-entrapment crystallization of corresponding phenocryst-host according to Danyushevsky and Plechov method using program Petrolog 3 [56]. This data treatment takes into consideration the diffusion exchange of FeO and MgO along the walls of the inclusion and the olivine host, then the Fe loss from the inclusion into the olivine host (Figure S6). Recalculated compositions from MIs in samples from HLN2A6 and HLN2A9 (Figure 3 and Table S7) were classified as basalts to basaltic andesites, and few andesites were observed ($SiO_2$ = 47.1–58.0 wt.% and $K_2O$ = 0.24–1.24 wt.%) and plotted in the field from calc-alkaline (CA) to high-potassium calc-alkaline (HKCA) rocks (Figure 3). Uncorrected values, including groundmass glasses, are available in the Supplementary Materials (Figure S6, Table S8). Corresponding olivine host crystals are listed in Table S9. The inclusion samples from HLN2A6 were richer in $K_2O$ (0.68–1.24 wt.%) and were classified mostly as basaltic andesites (*n* = 4 andesites), while olivine-hosted MIs from HLN2A9 were mostly basaltic (Figure 3), with lower contents of $K_2O$ (0.24–1.09 wt.%). The MIs were $SiO_2$-richer in HLN2A6 (52.6–58.0 wt.%) than in HLN2A9 (47.6–55.4 wt.%). Matrix glasses are basaltic andesites, having higher $K_2O$ (0.80–0.87 wt.%) and $SiO_2$ (53.0–53.7 wt.%) in HLN2A6 compared to HLN2A9, which contains $K_2O$ (0.74–0.82 wt.%) and $SiO_2$ (52.8–54.9 wt.%), respectively (Figure 3 and Table S7).

Corrected MI compositions (Figure 3 and Table S7) were processed by adjusting KD of olivine-melt and compared with groundmass and bulk rock compositions for $FeO_T/MgO$ ratio (Figure S6). In 12 MIs, important corrections were necessary (32% olivine, to a maximum of >10% olivine addition) [56]. The inclusions were preserved to moderate Mg# in the weakly primitive olivines ($Fo_{76-86}$); 30 of them [56] had an original $FeO_T$ of 0.2–7.3 below the corresponding host rock with $FeO_T$ vs. MgO trending and PEC results on MIs lowered the tendency at lower FeOT, while equilibrium was reached with Petrolog 3. The Fe-loss was corrected by using Petrolog 3 in addition to adjusting the iron content at equilibrium with MgO, considering the total iron content by assuming that $Fe^{2+} = \Sigma Fe$. The corrected melt compositions for NNO [57,58] had total iron species-consistent $Fe^{2+}/Fe^{3+}$ ratios according to the modeled compositions. The $Fe^{2+}/Fe^{3+}$ ratio of HLN2A6 (4.2–4.5) was similar to that of HLN2A9 (4.2–4.6) for modelling starting melt composition; despite this, the bulk rock titration method suggested $FeO/Fe_2O_3$ was lower in HLN2A6 (1.008) in comparison to HLN2A9 (4.540).

### 3.6. Volatile Contents

The sulfur contents (S) is reported in Table S7 and converted using $SO_3$ from Table S8. The results were obtained on the olivine-hosted MIs of *n* = 10 samples from HLN2A6 (51–568 ppm and mean 331 ± 153 ppm S) and *n* = 24 samples from HLN2A9 (41–2254 ppm and mean 940 ± 692 ppm S). The chlorine content (Cl) ranged between 27 and 953 ppm in HLN2A6 (mean 389 ± 306 ppm Cl) and was almost twice this amount, at 190–2126 ppm, in HLN2A9 (mean 591 ± 455 ppm Cl). The fluorine content ranged between 548 and 1449 ppm in HLN2A6 (mean 974 ± 322 ppm F) and between 62 and 1963 ppm in HLN2A9 (mean 975 ± 542 ppm F). As an internal reference for variation of volatile contents, a single olivine-hosted HLN2A9 crystal recorded ten MI for which the S, Cl, and F contents varied in the ranges 703–2254, 380–2126, and 493–1963 ppm, respectively.

Inconsistent extreme S values were recorded in sample # hln2a91mZX2 (2254 ppm S). A similarly excessive result for Cl occurred in this sample (2126 ppm Cl). In comparison, matrix glasses of HLN2A6 contained S 124–249 ppm, Cl 310–621 ppm, and F 425–564 ppm. HLN2A9 contained S 114–391 ppm, Cl 284–977 ppm, and F 906–1328 ppm (with sample hln2a96xr topping >4000 ppm F). If the extreme values are considered as outliers, the average volatile content in HLN2A9 exceeded that of HLN2A6 by 1.6, 2.8, and 1.5 for the

$CO_2$, S, and Cl contents, respectively, but both series of samples had consistent $H_2O$ and fluorine contents (ratio of ~1/1).

## 4. Discussion

### 4.1. Textural Relationships: Implications of Melt Inclusions and Clast Cooling Rates

Textural and compositional characteristics of the olivine-hosted MIs (Tables S3 and S7) testify on the variability of crystallization conditions from olivine phenocrysts (Figures 4 and 5; Table S7; Figures S3, S5 and S9). Such observations on MIs may reflect changes in cooling rates for the ascending cooling magma below LNG2 [14,15]. The groundmass microlite and phenocryst texture (Figure S4; Table S1) is used with matrix glass composition (Figures 3 and 5; Table S7), both as indicators of cooling rate variation from the residual melt during clast solidification [59–62]. The combination of such petrographic characteristics is compared among pyroclasts across the three eruptive centers (HLN2A6, HLN2A7, and HLN2A9), which complete qualitative observations on the vesicularity, crystallinity, shape, and texture of the sampled pyroclasts (Figure S1; Table S1).

The most degassed inclusions (<2.0 wt.% $H_2O$) appear to reflect the chemical behavior of water lost, as seen in Figure 5a, in which pressure changes are not necessarily related. After correcting for PEC, the sulfur and water distribution still followed the degassing patterns, but two populations of data mark a remarkable shift in the water content, which could be attributed to distinct cooling rates of the system [63]. On the one hand, this water loss was typical of post-entrapment processes in several of our samples below 2.0 wt.% $H_2O$ (Figure 5a–c). On the other hand, the chemistry of inclusions from lapilli scoria fragments differs from the bomb fragments sampled in this study (Figure 2). The results from Lloyd's work [63] were compared independently to the volcanic source (volcan Fuego, 1974; [64]), and we found that the clast size and cooling rate of clasts and inclusions could be closely related and therefore appear to be the main factor inducing water loss. This assumption is presented here as the MIs from the present work are in the composition range of the MIs (basalt to andesite) used to model the batch fractionation model used by Lloyd [63].

To verify if cooling rate has a direct effect at the scale of inclusions, the $K_2O$ and $H_2O$ content were compared to the inclusion volume in Figure 5c,d. We suspect PEC diffusion to affect its primitive melt composition [12–14,65]. The volume of solidified inclusions as observed in laboratory and the measured afterward polishing is probably indicative of larger surface contact area of MI versus olivine host [48,65]. If this association is correct, a decrease in inclusion size is related to PEC diffusion of major elements and re-equilibration of MI composition (red and blue color rectangle in Figure 5c,d), which also implies loss of water [56,65]. Otherwise, the quality of the glass surface signal from FTIR and EMP may be affected by artefacts that are not visible during analysis.

If the cooling rate is completely ruled out as an alternative explanation, differentiated MIs (rich $K_2O$ content) should be re-equilibrated with the olivine host [65,66] and preserve less water wt%. For this aim, the same elements ($K_2O$; $H_2O$) can be compared between groundmass and the inclusions to verify if the residual melt formed at the phenocryst rims is following cogenetic degassing trends. The trapped melt of the inclusion is supposed to be recording earlier stages of olivine growth [12,13] (Figure 5c,d). As for most of the inclusions, the $K_2O$ are less pronounced (magma is more primitive) and $H_2O$ is richer (less degassed), but many exceptions are observed for inclusions and some record lower water contents.

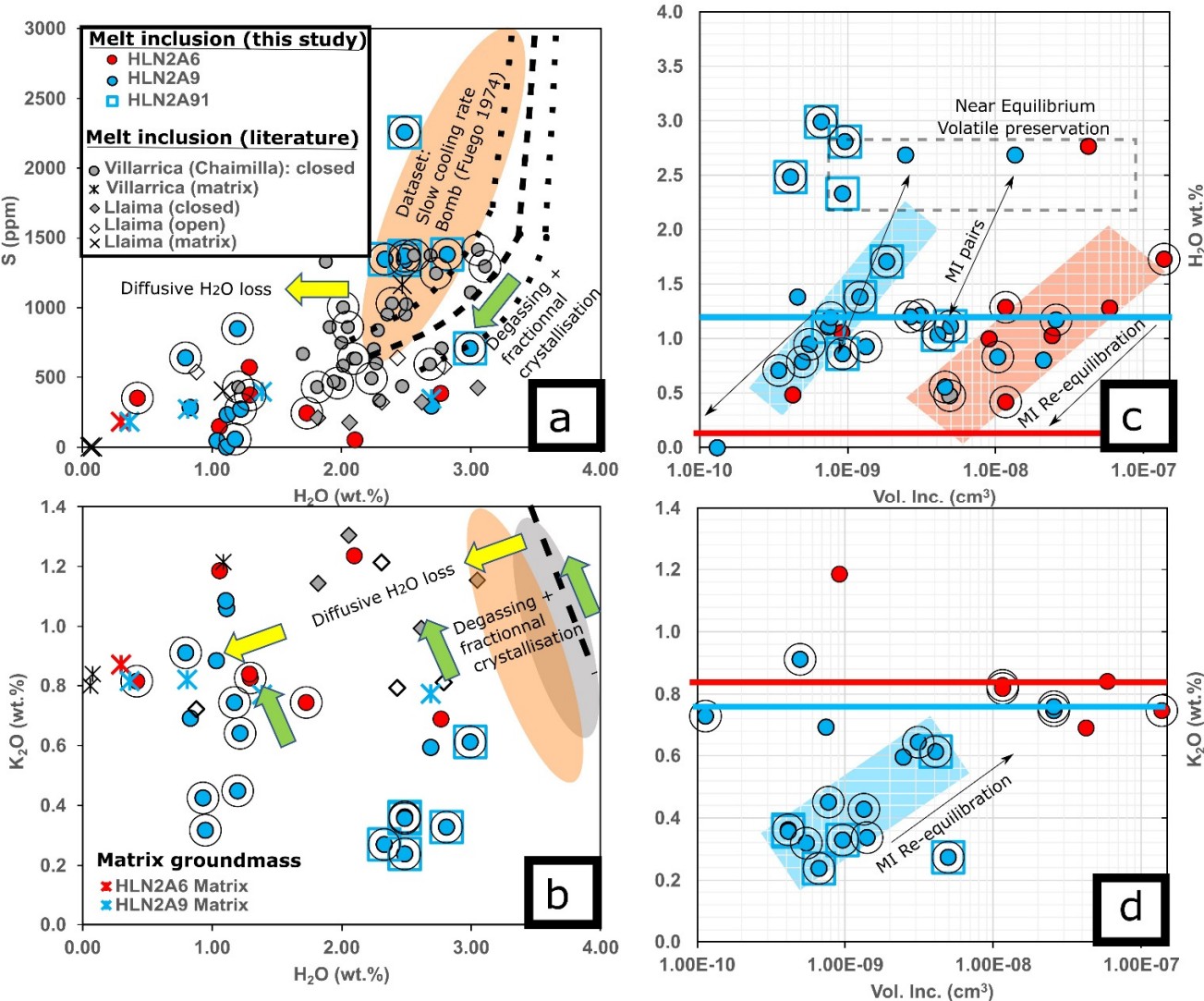

**Figure 5.** Factors controlling preservation of volatiles in LNG2 samples. The sample HLN2a91 has *n* = 10 data represented by a blue rim square. (**a**) $H_2O$ versus sulfur variation in MIs from this LNG2 highlighting the effect of diffusive water loss with yellow arrow. The solid dashed line represents the approximate limits for batch fractionation model used by Lloyd [63]. The degassing model tilted line represents an inclusion dataset of uncorrected ash and lapilli MIs from Lloyd et al. (2013) and its extension (dotted lines), while the reddish ellipse represents the inclusions from the bomb sample affected by water diffusion. All symbols with large transparent circles represent inclusions with an observed vapor bubble. Literature MIs data from Llaima volcano and Chaimilla event at Villarrica are mentioned in Supplementary Material Figure S6. (**b**) $K_2O$ versus $H_2O$ variation in MIs from LNG2 to distinguish between diffusive $H_2O$ loss (yellow arrow) and degassing + fractional crystallization (green arrow). Grey ellipse dataset of uncorrected ash and lapilli MIs from Lloyd et al. (2013). (**c**) $H_2O$ versus inclusion volume (Vol. Inc. in $cm^3$) variation in MIs. Matrix glass maximums $H_2O$ wt.% contents are represented by red (HLN2A6) and blue (HLN2A9) horizontal lines. Two end member arrows are linking two closed MIs pairs with clear distinction between $CO_2$ contents between the presence of an observed bubble (or not). (**d**) $K_2O$ wt.% versus inclusion volume in cubic centimeters (Vol. Inc. in $cm^3$).

If water contents are generally lower than in preserved MIs, it is not necessarily a surprise to observe richer amount in groundmass that are still not degassed. The same pattern is present in MIs, consequently lowering the $K_2O$ versus $H_2O$ content after the degassing processes; this mechanism has been suggested by Johnson, E. [62] to result after decompression. Within this order of idea, the crystallization of microlites in the groundmass is reflected in the variation of $K_2O$ content from groundmass, which could be

caused by cooling in the shallow storage region or by small reductions in the pressure of $H_2O$ (Figure 5b).

Despite this additional variable, PEC-MIs datasets generally follow the degassing trends found in literature (Llaima [67], Chaimilla event, Villarrica [28]). The presence of a positive shift in the water content for some groups of inclusions observed in the same olivine cannot be ignored (Figure 5a–c). As in the first instance, the quantity of glassy inclusions (20 versus 16%) and frequency of presence of a vapor bubble (1 versus 3%) in both HLN2A9 and HLN2A6F agree particularly well, but MIs from HLN2a6 demonstrate less water contents. If higher $H_2O$ loss is related to slow-cooling pyroclasts [63], such as the HLN2a6, or even by comparing effusive product such as lava [63,68,69], it is no exception that melt compositions are affected before being trapped in the olivine host, but also during post-entrapment processes when the whole system continues to cool down.

No inclusions were analyzed in clast HLN2A6A, but olivine populations represented stable conditions, such as in the case of few olivines with inclusions [14,15] or the lower rate of reabsorption (in olivine) and sieved textures (in plagioclases), which could indicate balanced conditions in the magmatic system [61]. Despite no conditions preserving many MIs, unlike HLN2A6A, HLN2A6F is a clast given a fusiform shape which indicates that plastic deformation was still occurring after fragmentation.

The groundmass texture also demonstrates distinct features; for example, the microlites of the groundmass in HLN2A6F are not oriented with respect to the vesicles, or to the phenocrysts (unlike HLN2A6A). The vesicles in HLN2A6F qualitatively show slight evidence of coalescence (Figure S4), perhaps the preservation of vesicularity is also weak [70]. The cooling rates for HLN2A6F would have been comparatively slower with respect to those evidenced for HLN2A6A since the clast suffered from plastic deformation rather than fragile rupture. The lower microlite content and larger size of the phenocrysts support this idea.

In the HLN2A6 eruptive center, there is high groundmass microlite density, particularly for plagioclases and moderate vesicularity (Table S1). The thin section and SEM petrographic characteristics of the HLN2A6A and HLN2A6F clasts (Figures S2 and S4, Table S1) testify for an increase in the cooling rate and a decrease in the growth rate of the crystals, allowing the nucleation of plagioclase microlites of the smallest size [59]. In plagioclase phenocrysts, the presence of frequent sieved textures and re-entrant/reabsorption edges could have been linked to depressurization during the magmatic ascent [61]. Of the three eruptive centers, HLN2A7 clast has the highest percentage of vesicles and the highest number of elongated vesicle morphologies (Table S1). It also represents an effusive center of emission between HLN2A9 and HLN2A6 with large olivine and plagioclase phenocrysts, but low microlite density. We stipulate the slow cooling rates of the effusive event at HLN2A7 and HLN2A8 must have played a role for preserving olivine-hosted inclusions. With respect to this idea, the olivine population shows clear dominance in recrystallized inclusions (e.g., G5), which result from slow quenching (Figures S3 and S5 and Table S3), and most of all in the few glassy/vitreous surfaces with conditions for realizing MI studies (e.g., G2+G3).

### 4.2. Degassing Mechanisms

The concentration of volatiles and saturation pressure were previously calculated according to the Iacono-Marziano's model [71], under the condition of minimum saturation of the magma in volatiles (Table S7). Such values represent the minimum pressure that could be affected by post-entrapment process mechanisms.

In some cases, the MI could be restored for $CO_2$ content [53,70–73] or other volatiles species (sulfur [74,75]) discounted in the presence of shrinkage bubbles, or even for the diffusion coefficient for water loss from the MI (e.g., [37,71]). In the present work, the quantification of shrinkage bubble $CO_2$ content has not been considered with Raman analysis (e.g., [37,53]), for which is the object of an additional MI study to determine if the "vapor" bubbles are "shrinkage" bubbles; this is required to conform to the criteria for this classification (e.g., [53]). Despite of this, to perform a full procedure of $CO_2$ correc-

tion, it is recommended experimental reheating (e.g., [72]), and test numerical modeling (e.g., [37,73]) with petrographic description of the bubble with high precision on its volume (vol.% bubble).

In accordance with the results obtained from the concentration of volatile species detailed in Table S7 (Figures 5 and 6), several scenarios support variation of the major volatile contents ($CO_2$, $H_2O$, S, Cl, F). To start with, a series of low $K_2O$ content values were identified in samples from HLN2A9 (Figures 4 and 5), mostly from the non-equilibrated MIs.

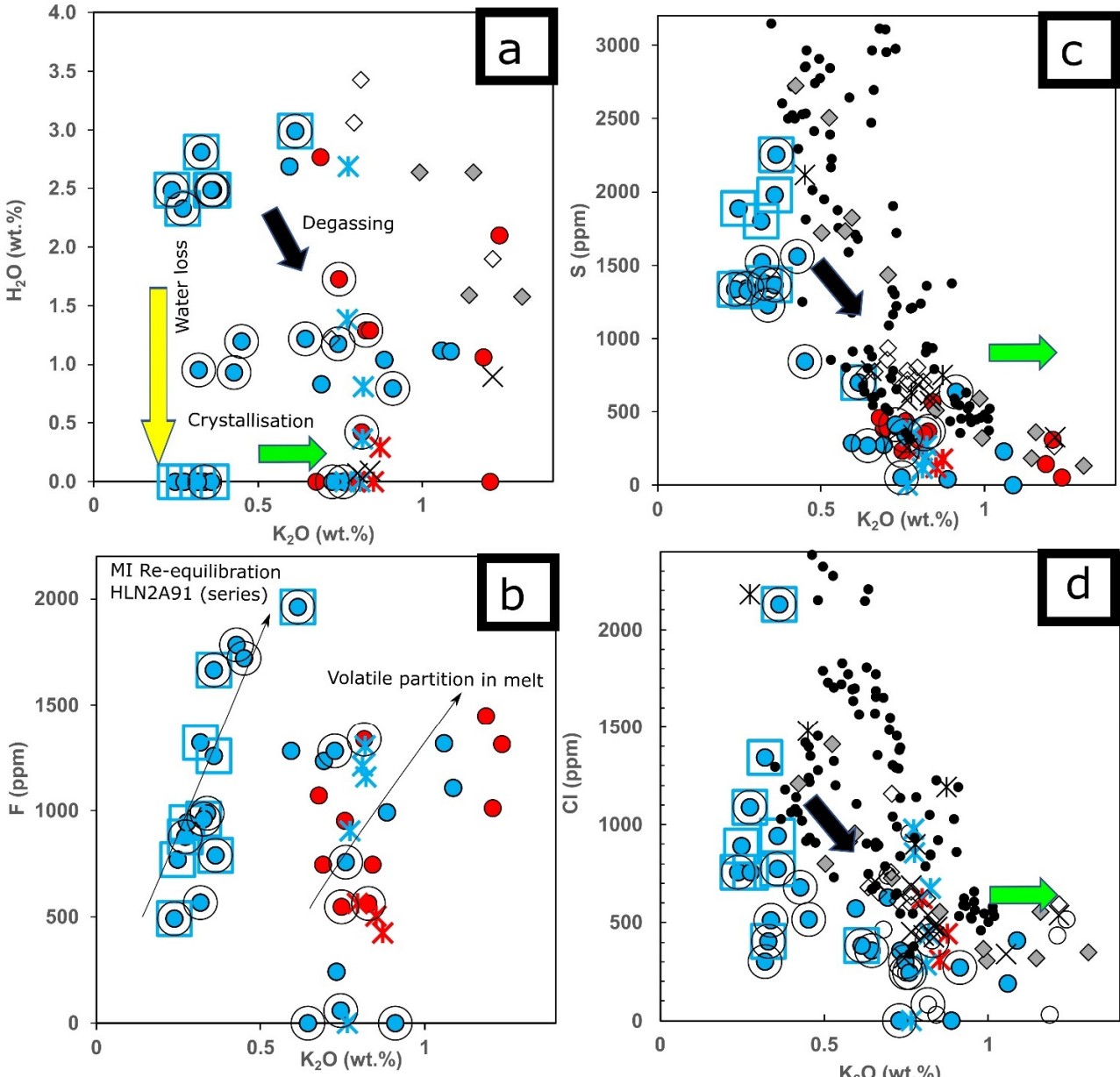

**Figure 6.** Variation of volatile contents in LNG2 samples. (**a**) $H_2O$ versus $K_2O$, (**b**) S versus $K_2O$, (**c**) Cl versus $K_2O$, and (**d**) F versus $K_2O$. Green arrow represents "crystallization", yellow arrow "water loss" ($H_2O$) and black arrow "degassing". Symbology is the same as previous figures. CVS stands for central volcanic zone and represent inclusions from [49].

The MIs compositions were compared to published datasets measured in Chaimilla deposits emitted from Villarrica central crater [28], the only eruption in the sector to be studied for this type of material and which correspond to mafic to intermediate MI compositions (Figure 3). Elsewhere, the Llaima volcano and collection of data from CVS are also taken as comparison with larger ranges of $SiO_2$ contents [49,67]. The CVS data

are compared to basaltic andesites from Los Nevados (LNG2) to verify if the degassing and crystallizing effect contribute to the same way on the MI content of major volatiles and oxides. This issue was not apparent in most samples from Villarrica (Chaimilla) and Llaima, which followed the degassing and crystallizing curves (e.g., clear variation with magma evolution [28,67]). For the rest of the LNG2 samples, fractional crystallization patterns clearly corresponded to the factor that influenced higher $K_2O$ contents (Figure 6). The most primitive MI compositions (low $K_2O$ of 0.24–0.55 wt.%) also matched the slightly higher forsterite content in samples from HLN2A9 ($Fo_{81-86}$). Consequently, the primitive ± less degassed melts are consistently trapped in primitive olivine crystals from LNG2, but the volatile contents of less differentiated MIs are well preserved in comparison to the MIs from Chaimilla [28]. The drastic shift to lower the volatiles contents could consequently reflect the decompression stage at lower $K_2O$ contents (green arrow in Figure 6).

The sulfur or chlorine contents of the sourced magmas were richer in some volcanic systems south of CVS (Cabeza de Vaca, Apagado), or alternatively, they suffered shallow conditions of mixing, which overprinted the primitive compositions. This mixing feature is not supported in the petrographic observations (Table S1 and Figure S1), or by other chemical trends in our samples. It is reasonable that chlorine solubility is decreased in the mafic melt which undergoes crystallization processes; it is incompatible, and Cl negatively correlates with $K_2O$ [76].

Halogens such as F have distinct signatures compared to Cl, as they do not behave the same way in both HLN2G samples (Figure 6b,d). The case of fluorine is particularly complex; its content variation observed along MI series from a single crystal (e.g. HLN2A91) may be explained by compositional MI re-equilibration or it belongs to various degrees of post entrapment crystallization occurring along MI-crystal rims (Figure 6b). Its general degassing pattern appears less clear like the other volatiles, except a general F enrichment with $K_2O$ in both inclusion group HLN2A9 and HLN2A6. The group of inclusions and matrix glasses from HLN2A6 is rich in $K_2O$, with a corresponding low Cl or F content, indicating the residual melt has already been degassed. For comparison, 3/35 samples had $F/K_2O$ ratios of >3.0 and the rest had <1.0, which is consistent with subduction-related basalts or undifferentiated end members [76,77]. According to Martini [78], fluorine species are more soluble than water in silicate melts, thus partition into the liquid melt phase rather than into the fluid during differentiation. To better test if fluorine partitions to the fluid or gas phase in the emissions of both materials, it is recommended to verify its content variation with refractory REE [79].

After considering factors that influence volatile loss (Figure 5), one model of the decompression of the magmatic system was proposed for the least degassed MI sample HLN2a922, representing the characteristics with no PEC-related textures (series of HLN2A92). For this reason, the maximum measured dissolved $H_2O$ and $CO_2$ contents in our MIs are used to calculate model degassing paths in Figure 7a, for which we assume vapor saturation during entrapment. The corresponding MI with highest volatile content ($H_2O$ = 2.69 wt.%, $CO_2$ = 1586 ppm) with primitive composition correspondence (Mg#0.70, 51.9 wt.% $SiO_2$, <0.6 wt.%$K_2O$) was selected considering it was trapped in a primitive olivine crystal matching high forsterite content ($Fo_{86}$). The Iacono-Marziano model initialized with inclusion HLN2A922, representing the first stage of the LNG2 eruption (Figure 2), for which the decreasing $H_2O$-$CO_2$ is calculated using the assumption of open- or closed-system degassing. The selected MI is without shrinkage bubble with an initial pressure fixed at 249 Mpa (at T = 1120 °C). We caution that the calculated entrapment pressures are considered as minimum results due to high probability for MIs $H^+$ and/or $H_2O$ diffusion (Section 4.1) and the presence of shrinkage bubbles in some MI from the same olivine host.

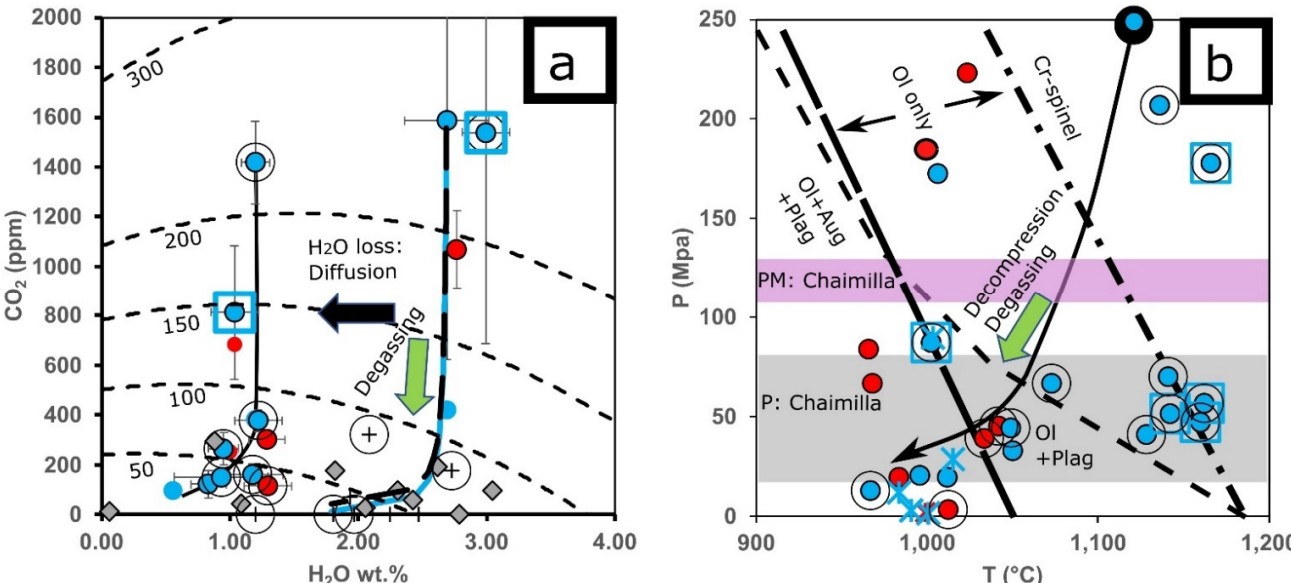

**Figure 7.** Pre-eruptive P-T conditions for inclusion entrapments. No effects are illustrated from post-entrapment modifications except for the formation of bubbles and $H_2O$ diffusion. (**a**) $CO_2$ versos $H_2O$ diagram. The solubility model uses [71] with MI sample HLN2a922. Only fully enclosed inclusions are shown (open circle for the presence of observed vapor bubble). Vapor saturation isobars are calculated using [71], considering MI without vapor bubble had a fluid water mole fraction of 0.249 (matching the $H_2O$-$CO_2$ content). The closed with open-system degassing path is illustrated with black dashed line and blue dashed line, respectively. Error bars from our study are determined in Section 3.4 (see also Figure S7). (**b**) Dry and hydrous phase relations for basaltic andesite are drawn approximatively and based on models of Moore and Carmichael (1998). Limit of mineral stability fields are illustrated with dotted lines (Ol is olivine; Aug is Augite; Plag is Plagioclase; Cr-Spinel is Chromium-Spinel). Only fully enclosed, glassy texture inclusions (open symbols) are plotted as they are considered to best represent the pre-eruptive melt with little to no effects from post-entrapment modifications. The green arrow represents the hypothetical magma crystallization path in tandem with decompression "degassing". The temperatures are from method of Putirka (2008). Symbology is the same as previous figures. Purple rectangle indicates range of pressures from MIs of Chaimilla event (PM: Chaimilla; 120 Mpa), grey rectangle are the ranges of pressure from Chaimilla for minimum entrapment pressure in MIs of Chaimilla event (P: Chaimilla; [28]).

The model demonstrated that either open- or closed-system degassing can fit for the data at a fluid water mole fraction of 0.249 (matching the $H_2O$-$CO_2$ contents from HLN2A922), because our dataset approximatively follows a vertical line between 2.6–3.0 wt.% $H_2O$. Despite several MI (*n* = 5) plotting along the closed degassing curve, the negative shift is clearly visible for water content for the rest of the samples, representing the effect of water loss by diffusion, as identified earlier (Figure 5). Some of the data of the $H_2O$ and $CO_2$ concentrations did not follow the trend of the theoretical model (Figure 7a); rather, the values lagged to the left, which would imply that an ideal theoretical behavior would not represent the data trend during entrapment of melts and olivine crystallization.

Saturation pressures were calculated on 26/39 MIs. For few other MIs, it was not possible to quantify and match all PEC-adjusted oxide contents of the corresponding glass to all the $CO_2$ contents. It is likely that, in some cases, the values of this volatile species were below the detection limit (bdl; Table S5), or that the spectrum of the crystal signal interfered with the glass signal from the inclusion. The pressure range calculated for HLN2A9 in 13 MIs varied between 12 and 248 Ma (Table S7 and Figure 7b), which would correspond to saturation depths between 0.5 and 9.0 km. HLN2A6 varied between 2.0 and 222 Mpa (0.1–8.1 km). These values are low and indicate that degassing processes occurred at the crustal levels. For comparison, the MIs recording minimum pressure for the Chaimilla eruptive event decrease from 91 to 37 Mpa according to FTIR data from [28]. The three eruptive centers studied at Los Nevados have the fact that the saturation depth occurs at superficial levels for similar $H_2O$-$CO_2$ ranges in common, from those deeper for

the maximum value of HLN2A6F (8.1 km) and the highest values of depth for HLN2A9, toward 9.0 km, but the shift in temperatures (Figure 7b) and water contents (Figure 7a) clearly indicate that HLN2a6 clasts contain certain MIs affected by diffusive loss.

On the map, HLN2A6 is about 100 m higher than HLN2A9 and diverges from it 500 m along the interpreted 45 °N fissure (Figure 2), which point out from a similar depth source for the plumbing system between the two emission sites. Despite sharing several data among the minimum pressures recorded for Chaimilla, the variation in solubility of the $H_2O$–$CO_2$ pairs show distinct pattern (Figure 7a,b). No evidence of common magma source is discussed here, but the $CO_2$ content measured at LNG2 is clearly preserved and variable in MIs, independently of the presence of observed bubbles.

Another similar trend in general terms corresponds to the fact that there is a high variability of the $CO_2$ content despite the two groups of $H_2O$ contents (Figures 5 and 7). $CO_2$ depends strongly on the pressure of the system; at lower pressures its solubility decreases, and it tends to segregate into a gaseous phase. This process, which depends on the solubility of volatile species, is consistent with the proposition that the olivine crystallizing, and inclusion entrapment processes occurred at a shallow depth.

### 4.3. Eruptive Style Variation

The deposit and pyroclastic fragments of the eruptive centers suggest that these originated in a Strombolian-style eruption [8]. However, the chronological variations of pyroclast textures and magma compositions between each center of emissions indicate that the eruptive dynamics were not constant during the eruption. According to mapping of the deposit (Figure 1c,d and Figure 2) and stratigraphic columns (rather lateral contacts and few vertical sections available; Figure S1), the eruptive centers represent series of cones for which deposits cumulated along the same fracture, from NE to SW. The HLN2a6 series cover the subsequent series of deposits from the other cones, with HLN2a7 covering HLN2a8 and HLN2a9 being surrounded by lava flows from the previous eruptive centers (HLN2a8 and HLN2a7) formed at higher altitudes. A relevant parameter when studying the variations in volatile contents with respect to different fragments is the limited information on the stratigraphy of the deposit. In the case of this work, the map produced by an additional campaign (Figure 2) clearly defined the chronology of each volcanic vent, from the oldest to the newest, as HLN2A9 > HLN2A7 > HLN2A6 (double-nested).

Previous mapping work [2,3] indicated that, in addition to the presence of scoria cones, there are also associated lavas continuing down the valley; their origin in the SW sector (Figure 1b) was confirmed by the photogrammetry results and mapping effort here as being sourced close to HLN2A8 vent (Figure 2). The unique field conditions provided quality orthomosaic pictures, with a close connection of the lava branch to the sector HLN2A7 and HLN2A8 vents (Figure 2). This suggests effusive activity, like observations of Cono Navidad on the flank of the Lonquimay stratovolcano [80].

The geometry of the deposit shows that the largest falling pyroclast fragments with "agglutinated" characteristics are found at the base of the sequence as levee (Figures S1 and S2, and Figure 2); therefore, they must correspond to early stages of the eruption. In other eruptive centers, such as HLN2A9, pyroclasts of this type are also observed in the middle zones and towards the upper sequences of the deposits, also indicating fluctuation in eruptive style. The presence of diagnostic fragments characteristic of a more effusive activity (spatter, agglutinations, and clasts of fluid morphology) would show variability in magma viscosity [9].

Vesicular clasts with sideromelane characteristics were possibly formed during Strombolian activity [81] with the occurrence of efficient magmatic degassing; however, it was not possible to apply a quantitative approach here (e.g., Pelagatos [82,83]), since it is not clear how the fine to coarse ashes or lapilli size tephra are conserved around the sector, to proceed with a sideromelane versus tachylite comparison [82]. Instead, detailed petrographic information on the sampled clasts and a textural study of the inclusion population, with their size versus water content (Figure 5c,d), helped us to discriminate the effects of cooling

rates on the sampled rocks (Section 4.1). For example, the denser clasts in HLN2A6 are contained in a few vesicular and rich microlite groundmasses, indicating the solidification and fragmentation of a relatively cooled and degassed magma, which is also supported by the shift in S, Cl, and F volatile contents registered in the MIs (Section 4.2). The clast cooling rate [63] did not show a direct correlation in this study with the cooling rate of the inclusion systems, because of our low variety of fragment sizes.

The density of MIs in HLN2A9 and the presence of anomalous inclusions in a single phenocryst (HLN2A91) also suggest a variety of compositions (Figures 4–6), despite our expectation of encountering homogenous conditions [84,85]. Physical appearance is often related to the MI composition [14,15,84,86], and we tested the reproducibility of results in this phenocryst and observed different textures in distinct sectors of the phenocrysts, which are not related to growth zones. As the textures are diverse and boundary effects strongly influence the $K_2O$ variation in the inclusion (Figures 4 and 5), it can be proposed that distinct sectors of HLN2A91 phenocryst did experience direct $H_2O$ loss and major element exchanges due to boundary effects with olivine hosts (proportional to inclusion size, Figures 4 and 5). The fraction of samples with G3 and G4 MI population on HLN2A91 border, as observed in other olivines, demonstrated frequent larger MIs (>50 μm larger axis) with dark coffee tone or dark thin border rim at the olivine-inclusion interface, indicating possible reaction of water with iron species involving water-loss; such features have been commented for MI that recorded water loss by diffusion [86,87].

According to observations by various authors of samples from volcanic arcs [86], numerous anomaly inclusions may be related to fast crystallization rates within plumbing systems, which occur close to the wall rock of shallow conduits (mush zone, [59,85]); this contrasts with the fact that olivine that develops inclusions necessarily relates to fast crystal growth, but in this case, a large difference exists between the center of the magma plumbing system and the relatively cool peripheral portions, where the cooling rate should be faster. The presence of numerous well-preserved concentric small inclusions (with G2+G3 being numerous, Table S3 and Figure 4a,b) is particularly notable for the HLN2A9 series in comparison to the HLN2A6 series (Table S3), which could be interpreted as an initial close contact of phenocrysts with the cold plumbing system opening [86], while at the end of the magma circulation, olivine crystals did not develop such textures [14,15].

As was proposed by Pastén [31], for the three eruptive centers studied (HLN2A6, HLN2A9, and HLN2A7), the degassing mechanism that occurred during the LGN2 eruption had the potential to fluctuate and affect the explosivity at a single vent. The scenario was simplified [6–8] and adapted to the HLN site in Figure 8. More specifically, it typically occurs due to the release of the magmatic volatiles contained in the melt through a "slug" type of flow regime, driven by the rise and bursting of bubbles [88], for which two models have been proposed: (1) the so-called "dependent on the ascent speed" and (2) the "foam collapse".

Furthermore, the interpreted lower cooling rate of sampled clast (Figure 5) is also observed with a lower evidence of coalescence between bubbles in this sample (Table S1), a variation in the late-degassing process could occur with respect to the other eruptive centers. Comparatively, HLN2A7 and HLN2A9 represent more vesicular pyroclasts with greater evidence of coalescence, while HLN2A6 is made up of fewer vesicular and more crystalline fragments (Figure S4). Considering the differentiated composition (reaching MI andesite classification; Figure 3), slower quenching rates for olivine-hosted MIs, the mentioned clast (micro) textural evidence (Figures S1 and S2, Table S1), and finally the frequent presence of similar bomb-size crystalline clasts around the vent (Figure 2), the eruptive center HLN2A6 may represent viscous magmas leading to late explosive processes compared to HLN2A9 and HLN2A7 (Figure 2).

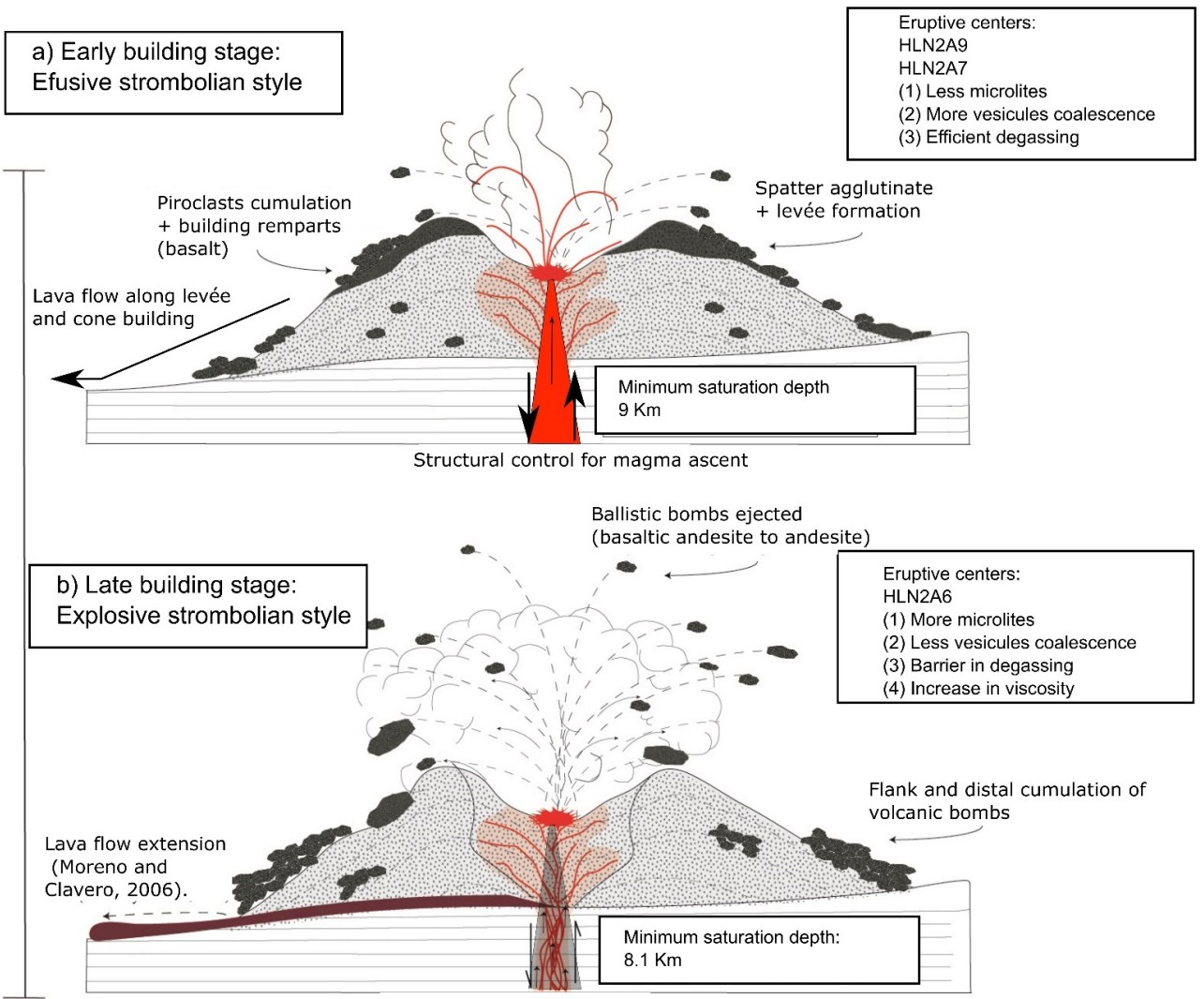

**Figure 8.** Modeled eruptive stage of Los Nevados Subgroup 2.

## 5. Conclusions

The petrological characteristics of the scoriaceous clasts analyzed in this paper, and of the deposits of the three minor eruptive centers associated with the Los Nevados Subgroup 2 of the Los Nevados Group, suggest variations in the eruptive dynamics of the erupted magmas. The characteristics of the deposits indicate that the eruptive style was Strombolian, and the spatial characteristics of the emission centers suggest that the magmatic location is related to the structural system of the area—mainly to the fractures of the N60 °E preferential orientation. The spatial variations of the deposits of the eruptive centers suggest that the transitions in the Strombolian eruptive style could have occurred both temporally and spatially (Figure 8). The process of construction of the cones was complex, and two stages could be characterized, beginning downhill at the NE end, but ending on top of a hill at the SW eruptive center located close to the Caldera 1 from Villarrica (Figures 1d and 2).

Initially, the eruptive process is interpreted to be rather effusive within the context of the Strombolian style. This stage was characterized by relatively slow magmatic ascent rates around 249 Mpa (9.0 km) that allowed for the growth of phenocrysts and the coalescence of bubbles, which favored the pre-eruptive degassing of the system and facilitated the first fragmentation processes. Extrusion of spatter and other pyroclastic fragments accumulated in the flanks of the emission centers along the N60 °E fracture. Evidence of this is found in

the basal deposits of the eruptive center HLN2A6 and, to a greater extent, in the deposits of the minor eruptive centers HLN2A9 and HLN2A7, in which the clasts represent layers of spatter deposits that changes to layers of scoria deposit. The petrography of the sampled clasts showed increasing density of micro and macro vesicles. This temporal evolution in the type of clast (vesicular scoria rather than spatter) would be interesting to quantify, but somehow the behavior of the gas phase of the system determines to a large extent what the eruptive dynamics will be and therefore what type of clast is emitted. In turn, the textural study of MIs in HLN2A9 supports, slow ascent conditions at the beginning and fast cooling rates for the batch of crystallizing and rising magma through new "cold" fracture systems.

Later, a second stage had more explosive characteristics (also within the Strombolian eruptive style range) and thus allowed the extrusion of scoriaceous and dense pyroclastic fragments of MIs basaltic andesite to andesitic compositions. It is proposed that the depressurization of the magmatic system (222 to <12 Mpa) caused degassing at the surface level (lithostatic depths of 8.1 to 0.1 km), and the crystallization of microlites in groundmass below 90 Mpa (<3.0 km) favored the degassing of the volatiles, together with relatively slower ascent rates. The degassing of the system also allowed the crystallization of microlites of plagioclase, then larger differentiation of the melt preserved in MIs at the last emitting vent (andesitic compositions), which consequently increased the viscosity and favored the explosiveness of the system at specific volcanic vents along the initial fracture (vent HLN2A6).

**Supplementary Materials:** The following are available online at https://www.mdpi.com/article/10.3390/geosciences11080309/s1: Figure S1. Field photography of sampling sites and schematic stratigraphic columns. Figure S2. Backscattered Electron images. Figure S3. Melt inclusion populations in olivine. Figure S4. Pictures from petrographic thin sections. Figure S5. Pictures from prepared olivine wafers. Figure S6. Post entrapment crystallization corrections of olivine-hosted melt inclusions. Figure S7. Microanalysis of melt inclusions: preparation and data processing. Table S1. Petrographic description of thin sections. Table S2. Bulk rock analysis Table S3. Textural characteristics of inclusions at the eruptive centers HLN2A9, HLN2A6, and HLN2A7. Table S4. Glass standard duplicates, EMPA 2019 session. Table S5. $H_2O$-$CO_2$ contents from FTIR. Table S6. Determination of carbon dioxide content with FTIR. Table S7. Volatiles and major elements in olivine-hosted melt inclusions and glass rims. Table S8. Chemistry of glass surfaces by EMPA. Table S9. Chemistry of olivine minerals by EMPA. Supplementary material Video S1. Drone flight survey of the Los Nevados Subgroup 2: SW Sector.

**Author Contributions:** Conceptualization, P.R. and D.P. (Daniela Pasten); methodology, P.R., G.L., D.P. (Daniela Pasten), G.D. and D.P. (Dante Paredes); software, P.R., G.D. and D.P. (Dante Paredes); validation, P.R., G.L. and D.P. (Daniela Pasten); formal analysis, P.R., D.P. (Daniela Pasten), G.D. and D.P. (Dante Paredes); investigation, P.R. and D.P. (Daniela Pasten); resources, P.R. and G.L.; data curation, P.R., D.P. (Daniela Pasten), G.D. and D.P. (Dante Paredes); writing—original draft preparation, P.R., D.P. (Daniela Pasten) and G.L.; writing—review and editing, P.R., D.P. (Daniela Pasten) and G.L.; visualization, P.R., G.L., D.P. (Daniela Pasten), G.D. and D.P. (Dante Paredes); supervision, P.R.; project administration, P.R.; funding acquisition, P.R. All authors have read and agreed to the published version of the manuscript.

**Funding:** This research was funded by (ANID) Fondecyt Iniciacion a la Investigacion de Philippe Robidoux, Etapa 2020, grant number N11190816. Additional resources were obtained via Start Up from Vicerrectoría de Investigación—Universidad Mayor (2017–2019).

**Institutional Review Board Statement:** Not applicable.

**Informed Consent Statement:** Nor applicable.

**Data Availability Statement:** Digital elevation model (DEM) of 12.5 m resolution (DEM ALOS-PALSAR; https://vertex.daac.asf.alaska.edu, accessed on 11 December 2018). This raster image corresponds to Data available in a publicly accessible repository which hold the specific doi:10.5067/Z97HFCNKR6VA.

**Acknowledgments:** Philippe Robidoux would like to acknowledge the ANID-FONDAP Project 15090013 and 15200001 granted to the Andean Geothermal Centre of Excellence (CEGA), as well as to Piergiorgio Scarlato and Manuela Nazzari, for allowing access to the HPHT laboratory of INGV in Rome, Italy, for electron microprobe analysis. Special thanks are given to Marina Vega (Laboratorio de Fluidos Corticales, UNAM, Juriquilla, Mexico) for helping with the SEM and FTIR analysis. Special recognition is given to former graduate students Aaron Sancho (U. Mayor), Gianni Russo Melchiori (alpine guide, Pucón) who assisted with the field rock sampling and finally Alberto Espinoza (Transmark), who processed the photogrammetry images.

**Conflicts of Interest:** The authors declare no conflict of interest. The funders had no role in the design of the study; in the collection, analyses, or interpretation of data; in the writing of the manuscript, or in the decision to publish the results.

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
