# Peer review of "Volatile Content Implications of Increasing Explosivity of the Strombolian Eruptive Style along the Fracture Opening on the NE Villarrica Flank: Minor Eruptive Centers in the Los Nevados Group 2"

_geosciences, doi:10.3390/geosciences11080309_

Round 1

Reviewer 1 Report

The paper deals with the description of petrology and composition of deposits from a flank scoria cone eruption(s) of Villarrica volcano. The authors present data on glass matrix, glass inclusions and mineral composition from scoria of different sizes (from lapilli to bomb); then they interpret the data based on the stratigraphic sequence and eruption reconstruction. Interpretation is done in terms of eruption and magma rise dynamics.

The paper requires some work before being published. I found several bias in data interpretation which should be corrected. I will detail my comments below.

The authors should correct words in technical english but also check grammar- I advise them to check terms.. for example  : 'microlite' and not 'microlith', top of the deposit and not roof 'or 'ceiling', the crater is covered and not topped; 'microphenocrysts' and not 'microphenocrystals'; more differentiated and not 'higher differentiated', 'escarpment' and not 'escarp'. The magma plumbing system has peripheral portions/periphery and not 'borders', 'scoria' or 'cinder cones 'and not 'pyroclast' or 'pyroclastic cones'. 'Clast' is a specific geological term and this preferable to 'fragment'; 'Groundmass' and not' fundamental mass', 'Tachylite' and not 'trachylite'; 'conduit' and not 'conduct'; 'dynamics' and not dynamism...

The paper is well structured, the introduction is informative, but the authors are not convincing when they justify the choice of peripheral eruptions at Villarrica to understand the role of degassing in controlling the eruptive dynamics. I suggest rewriting the corresponding paragraph.

Methods are explained in detail. Data are presented in detail, but more basic plots  are required to help the reader follow the text.

Interpretation is not always supported by data; I suggest major revision. Please see further comments for more details.

Line 55-58 Not clear. It should be rewritten.

Line 76-8. Do you mean that the scoria cones and lava associated to the studied eruption cover reworked tephra from Villarica and LNG2? and that the cones are made by scoria with variable textures? a vent is just an opening through which magma is emitted. Rephrase

line 88 Do you mean that you studied pressure dependent volatile equilibria such as dissolved H2O-CO2 ratio?

line 89. What is the lithological spatial limit? Do you mean that you correlated and defined the stratigraphic succession to define the chronology of the volcanic activity?

line 146. Is villarrica Unit 3 basal unit the Chaimilla eruption? if so, it should be clearly written. It is weird to associate a precise age to a large scale unit.

line 154-5 do you mean that the cones are horseshoe-shaped?

line 192. What is a transparent section?

line 217 what are small to medium size lapilli?

line 284. What is a levee deposit? the term levee is used to describe self confining structures either in lava flows or PDC deposits. Do you mean a rampart?

line 287 (and elsewhere) is % indicating vol %?

line 327. What is a rounded mound indicating? why pyroclasts accumulated forming a mound?

line 365. Incomplete sentence ?

line 383. What is this specimen used for? test repeatability of the measurements/error/variability of water content? the measured range is very high .

line 447-8 You can't discuss variability of parameters you did not measure, such as textural parameters, vesicularity and morphology. The variability of melt inclusion do reflect variability of crystallization conditions of the crystals that host them.

lines 454-460. Phenocrysts form in the magma reservoir, while microlites form either during eruption or after fragmentation (in the conduit, fountain or the deposit). Neither CSDs and Plag compositions were not measured in this work, so there is no basis for a model of multistage crystallization.

line 465. Glass inclusions are olivine hosted, so they reflect crystaalization conditions of olivine, not plagioclase.

line 506. Please define what you mean with pre-entrapment processes

line 512. The bomb shape is acquired in the fountain after fragmentation. When cooling becomes significant, the clast does not deform anymore plastically but becomes fragile.

line 529. The variation in textures are not linear within the same/similar eruptive style (strombolian/Hawaiian). Scoria vesicularity is the  product of several processes, which start from bubble nucleation during magma rise to postfragmentation degassing. Vesicularity itself (i.e. the volume concentration of bubbles) is the most tricky parameter, because it is heavily controlled by bubble expansion, which for this dynamics, is relevant at postfragmentation stage. More indicative (but still tricky, because they evolve not linearly) textural parameters such as BND, BSD, degree of coalescence, have not been measured in this work; there are no data for these interpretation. Secondly, olivine-hosted glass inclusions reflect volatile content at very deep stages of magma rise, before shallow degassing processes which are more important for eruption style.

Line 547. It is not clear to me why data are compared with LLaima  and CVS magma (and not with other published Villarica eruptions) Do the authors think that the magma feeding these volcanoes is the same? are they stored in the same system? Why should they follow the same liquid line of descent?

line 660-686 There are no evidences in this work to support any model on eruption style- no stratigraphy, no detailed study on deposit are presented. If the authors wish to do so, they should present actual stratigraphic data.

FIg. 1. Mark active summit vents of Villarrica volcano. What are the read line defining?

Fig. 2.Isoplets are isolines of equal maximum clasts. Do you mean that the lines define the spatial distribution of bomb clasts? The map is confusing.. HLN2A8b is mapped more like a flow than fallout /ballistic scoria. The boundary among units suggest that the Plv1 covers the cones in their lower portions and not viceversa; the shape of HLN2A6b cone suggest important erosion and coverage by Plv1 unit.

HLN2A7a and C units are swapped?

fig. 5. What is the meaning of the bomb from the 1974 Fuego eruption in these plot? do you think that the magma had the same composition? Data are bizarre: glass inclusions display the same range of water content of groundmass glass. this suggest no water loss. How is this explained?

fig. 6. Why this figure has a creamy background?

fig. 7. fig.a is one of the most relevant to the study and should be presented before in the work. How was the mineral stability calculated in the fig. b? Is it realistic? Is there opx in the scoria?

Author Response

Dear Reviewer,

Reviewer 2 Report

(Comments in attached PDF document)

Author Response

Dear Reviewer,

Reviewer 3 Report

This manuscript is a very well written article, with a big number of data, well worked with a number of analysis enough to support the discussions and conclusions presented in the text. I have only minor suggestions about the main text, which I provided as a pdf with annotations.

Author Response

Dear Reviewer,

Round 2

Reviewer 1 Report

I have no major comments

Reviewer 2 Report

I'm happy with the authors' responses to my prior comments, and recommend that the manuscript be published.